# Yttrium and Lithium Complexes with Diamidophosphane Ligand Bearing 2,1,3-Benzothiazolyl Substituent: Polydentate Complexation and Reversible NH–PH Tautomery

**Radmir M. Khisamov, Taisiya S. Sukhikh \*** [ID]**, Sergey N. Konchenko and Nikolay A. Pushkarevsky \*** [ID]

Nikolaev Institute of Inorganic Chemistry SB RAS, Akademika Lavrentieva Ave. 3, 630090 Novosibirsk, Russia
\* Correspondence: sukhikh@niic.nsc.ru (T.S.S.); nikolay@niic.nsc.ru (N.A.P.)

**Abstract:** Deprotonation of a bis(amino)phosphane $H_2L$ = PhP(HNBtd)$_2$ bearing a heterocyclic Btd = 2,1,3-benzothiadiazol-4-yl substituents at nitrogen atoms by silylamides LiNTms$_2$ and Y(NTms$_2$)$_3$ (Tms = trimethylsilylamide) results in lithium and yttrium complexes with the deprotonated HL$^-$ and L$^{2-}$ forms as $\kappa^2$-N and $\kappa^4$-N chelating ligands. A binuclear complex [LiHL]$_2$ was crystallized from Et$_2$O, and was shown to reversibly dissociate in thf (tetrahydrofuran) with the NH$_{(soln)}$–PH$_{(crystal)}$ tautomeric shift; the compound [Li$_2$L] was spectroscopically characterized. Yttrium readily forms stable bis-ligand complexes [YL$_2$]$^-$ and [YL(HL)]. In the latter, the H atom in HL resides on phosphorus; the coordination sphere remains accessible to another ligands, and it was crystallized as [{YL(HL)}$_2$(μ-dioxane)] species (YN$_8$O coordination). In the former complex, the coordination sphere was saturated (YN$_8$) by closer bound ligands; it was crystallized as a salt with [Li(thf)$_4$]$^+$. The mono-ligand complex could not be cleanly obtained in a 1:1 reaction of $H_2L$ and Y(NTms$_2$)$_3$, and was only crystallographically characterized as a dimer [YL(NTms)$_2$]$_2$. Partial oxidation of the central P atom with the formation of phosphine-oxide ligands PhP(O)(NBtd)$^{2-}$ was observed. They co-crystallize in the same position as non-oxidized ligands in [YL$_2$]$^-$ and [YL(NTms$_2$)]$_2$ species and participate in bonding between two units in the latter. TD-DFT calculations reveal that main transitions in the visible region of electronic spectra correspond to the charge transfer bands mostly associated with the orbitals located on Btd fragments.

**Keywords:** rare earth elements; coordination compounds; proton transfer; single crystal X-ray diffraction; UV-Vis spectroscopy; DFT calculations

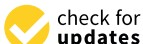



## 1. Introduction

Nitrogen–donor ligands are an important part of rare earth (Ln) coordination chemistry, as the nitrogen atoms can be combined into complex systems and simultaneously bear terminal or bridging functional substituents. A large variety of chelating N,N'-donor chelating ligands are implemented for stabilization, and their bi- or polydentate coordination ensures sufficient stability of Ln complexes. The donor N atoms can be joined by various types of bridges. In the simplest case, there is only one atom between a pair of N-donor atoms (Scheme 1); corresponding ligands, such as amidinates and guanidinates, have become extremely popular in Ln chemistry [1–7]. Despite the structural simplicity, there are fewer examples of the Ln complexes with the other central atoms (E) in the ligands. The most available alternative is the Si-centred ligands (56 examples of Ln complexes in the CCDC [8]), which can be readily prepared and functionalized [9–17]. Of the other main group elements, which form sufficiently strong E–N bonds, the ligands with B, N, P, and S in the position of the central atom (17, 16, 46, and 6 examples of Ln complexes in the CCDC, respectively [8]) seem to be the most accessible and versatile. The other main group elements (e.g., those from 4th and lower periods) should form weaker E–N bonds, and such ligands should be even less stable.

**Scheme 1.** *Left:* Common formula of N,N′-chelating ligands with one bridging atom E. *Top:* possible phosphorus-centred variants: with pentavalent P (type **A**) and trivalent P atoms (types **B–D**). *Bottom:* the bis(amido)phosphane proligand used in this work with its schematic depiction.

Among the above-mentioned possibilities, those with E = Si, P, and S have longer E–N bonds as compared to the elements of the second period (B, N), and thus form larger bite angles at the central atom (N–Ln–N), more suitable for large ionic radii of the rare earth cations. Phosphorus, however, has several advantages: it provides a ready NMR registration, and exists in two stable redox states (3+ and 5+), thus leading to a variety of possible ligands (Scheme 1), whose chemistry can be monitored in solutions. Rare earth complexes with P-centred N,N′-donor ligands have been known for a while [18,19], and some of them were successfully employed as polymerization catalysts [20–24]. Still, it is interesting to note that only complexes with iminophosphonamide ligands (i.e., those with pentavalent phosphorus in the central position, type A on Scheme 1) have been studied. The ligands with trivalent phosphorus (types B,C,D) have never been involved in the lanthanide chemistry, although they have been used in transition metal chemistry to some extent.

Recently, we have explored the chemistry of rare earth complexes with dianionic silanediamido ($R'_2Si(NR)_2^{2-}$) ligands, which have proven to sufficiently stabilize metal centres and provide sensitization to luminescence of some of them [17]. It was of further interest to see how the analogous bis(amido)phosphane(III) ligands (type B) will support the rare earth cations. Complexes of main group metals (Al, Mg) with such ligands have been recently explored [25,26]. In this work, we aimed to study the complexation with a bis(amido)phosphane ligand, $PhP(NHBtd)_2$ ($H_2L$), in its NH-deprotonated forms ($HL^-$ and $L^{2-}$), where Btd is a flat 2,1,3-benzothiadiazol-4-yl substituent. This ligand was introduced in our earlier work on Zn and Cu complexes; [27] it contains an additional N atom in the thiadiazole ring, capable of coordination, so each N-Btd fragment can be chelating. Btd units are known as potent chromophore groups, mainly implemented as a part of various light-harvesting [28,29] or luminescent systems [30–32]. To study possible ways of complexation, we chose the $Y^{3+}$ cation, since it, being chemically and structurally close to the lanthanides from the middle of the row, also does not complicate NMR characterization of complexes.

## 2. Results and Discussion

### 2.1. Synthesis and Structures of Li Salts

It is known that bis-aminophosphanes $R'P(NHR)_2$ are readily deprotonated by strong bases, such as alkali or alkaline earth metal alkyls ($Li^nBu$, $Mg^nBu_2$, $NaCH_2Ph$) [25,33,34]. The obtained salts of active metals could be further used to substitute another anion (e.g., halide) in the coordination sphere of a rare earth cation during a salt metathesis reaction. On the other hand, the protonated form could be directly used in the reaction with a rare earth complex, where the ligands are represented by anions of weak acids, for example, amido-

or alkyl complexes. These anionic ligands are protonated and leave the coordination sphere as stable by-products. Earlier we obtained a Zn complex with the doubly-deprotonated $H_2L$ (i.e., $L^{2-}$) ligand in the reaction with the potassium bis(trimethylsilyl)amide ($KNTms_2$) as the base [27]. Hence, in this work we used a similar approach—the reactions of $H_2L$ with readily available lithium and yttrium amides, $LiNTms_2$ and $Y(NTms_2)_3$. We expected that the only by-product, amine $HNTms_2$, is well soluble and could be easily separated, as opposed to precipitates of alkali halides formed in salt metathesis reactions.

To check the availability of anionic forms of $H_2L$, we tried its deprotonation with $LiNTms_2$. In the reaction with one equivalent of $LiNTms_2$ in tetrahydrofuran (thf), the colour was instantly changed to dark red, and upon the concentration and addition of ether, the lithium complex $[Li(HL)]_2 \cdot Et_2O$ (**2**$_2 \cdot Et_2O$, Scheme 2) readily crystallizes in the form of lighter orange blocks. Hereinafter, a molecule or ion pair is denoted by a number (e.g., **2**), while the subscript denotes a dimeric species (e.g., **2**$_2$). An additional donor directly bound to a molecule is specified in brackets (e.g., **2**(thf)), while for a crystalline phase comprising a solvate molecule, the latter is specified after the dot (e.g., **2**$_2 \cdot Et_2O$).

**Scheme 2.** Synthesis of **2**$_2$ and its dissociation in solution.

According to the single-crystal XRD analysis, the molecule of **2**$_2$ has a binuclear structure with two equivalent Li central atoms related by two-fold rotation axis (Figure 1a). The ligand coordinates two metals via two pairs of N atoms in a chelate-bridging mode. Thus, the coordination polyhedron of Li is a tetrahedron {$N_4$}. The flat Btd and Ph fragments of different ligands form three pairs of π-stacks. One of them is formed by two Btd fragments positioned in antiparallel manner with the interplanar distance of 3.43 Å (Figure 1a). Another two stacks are formed between the Ph cycle of one ligand and one Btd cycle (with its thiadiazole part) of another ligand with the interplanar distance of ca. 3.5 Å (Figure 1b). The same molecular geometry is observed for the binuclear Zn complex, $[ZnL]_2$, which comprises tetrahedral $Zn^{2+}$ and dianionic $L^{2-}$ [27]. Since in **2**$_2$ the N atoms are engaged into short bonds with the Li, they cannot bear the remaining H atoms by steric reasons. The residual electron density map (Figure S1, Supporting information) indicates that the proton is located on the P atom, although this is not a solid argument due to the presence of some artefacts around the P atom arising from a slight disorder and/or somewhat low quality of the XRD data. A more reliable argument consists of shortened P–N bonds in **2**$_2$ as compared to $[ZnL_2]$ (1.60 Å vs. 1.71 Å). Moreover, there is a distinct peak at 2269 cm$^{-1}$ in the IR spectrum of this compound, corresponding to ν(PH) stretching. This peak is absent in the spectrum of $H_2L$. Instead, in its spectrum, a sharp peak related to ν(NH) stretching at 3340 cm$^{-1}$ is present. Thus, we conclude that in the solid **2**$_2 \cdot Et_2O$, the $HL^-$ anion comprises the proton at the P atom.

The notable colour change that occurs upon dissolution of **2**$_2$ (from orange in solid to dark red in thf solution; see also the discussion in Section 2.3) is most likely explained by solvation and dissociation of the dimeric complex. Interestingly, a characteristic doublet from a P–H bond was not observed in $^{31}P$ NMR spectrum (thf-$d_8$), which suggests that the acidic proton resides on the N atom in the dissolved complex. Thus, the complex represents a reversible NH to PH tautomeric transfer (Scheme 2). Previously, only a direct transformation of NH to PH form has been observed for anionic diaminophosphane ligands

in complexes of $Al^{3+}$ and $Mg^{2+}$ [25,26], although equilibrium between these two forms was observed for bis(alkylamido)phosphanes [35]. In $^1H$ spectra, the signals corresponding to Btd protons are very broad, which is possibly explained by labile coordination of $Li^+$ cations by the NBtd units in thf solution.

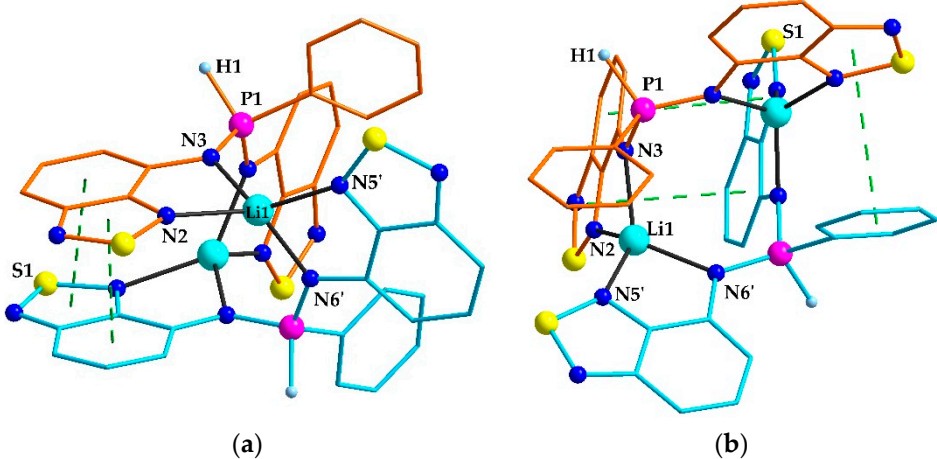

**Figure 1.** (**a**,**b**) Two projections of the structure of $2_2 \cdot Et_2O$ according to XRD. Solvate ether molecules and hydrogen atoms, except for those at the P atoms, are omitted. Different $HL^-$ ligands are coloured with orange and blue. Green dashed lines indicate $\pi$-$\pi$ interactions. Selected distances (Å): N2–Li1 2.253(8); N3–Li1 2.030(9); N5–Li11 2.111(8); N6–Li11 2.065(8). Symmetry code for ('): 1–*x*, *y*, 0.5–*z*.

The reaction of two equivalents of $LiNTms_2$ with $H_2L$ (**1**) in thf solution instantly leads to formation of a dark-blue mixture. The solid compound was separated after evaporation and washing with diethyl ether. Still, the $^{31}P$ NMR spectrum of this solid (**3**) contains two signals in the ratio of approximately 1:2 (at 77 and 70 ppm, respectively), which is at an expectedly lower field as compared to **1** or **2**. Although the reason for two distinct $^{31}P$ signals is not clear, similar behaviour was well documented for other dilithiated di(alkylamino)phosphanes [34], and likely corresponds to several oligomeric species in solutions [36]. In $^1H$ spectrum, all signals from the compound are broadened (those in aromatic region), which could also reflect speciation in solution. There are also signals corresponding to thf and $Et_2O$, which could be coordinated to $Li^+$ in the solid; if the corresponding amounts of these solvents are accounted for, the elemental analysis data correspond well to the theory. The bands of $\nu(NH)$ or $\nu(PH)$ stretching are expectedly absent in the IR spectrum of **3**. Despite the efforts to crystallize this compound from thf-ether mixtures, or with the addition of tetramethylethylenediamine (tmeda) or dimethoxyethane (dme) to chelate $Li^+$ cations, we could only observe slowly growing crystals in some cases in the form of thin plates, which were too small for a single-crystal XRD experiment.

## 2.2. Synthesis and Structures of Y Complexes

First reactions of $H_2L$ with $Y(NTms_2)_3$ were tried with a 1:1 stoichiometry, with an intent to substitute two amides with one $L^{2-}$, thus leaving one amide ligand available for further reactions (Scheme 3). The synthesis was conducted either by mixing two reagents simultaneously in cold solvent (thf, ether, or toluene), or upon slow addition of $H_2L$ into the solution of Y amide (local excess of the latter). It was noted that thf as the solvent notably slowed down the reaction. Its progress, observed by the darkening of the mixture, was slow at RT, and the process was visibly finished only after some heating (ca. 60 °C for several hours). On the contrary, the reaction in toluene or $Et_2O$ proceeded almost instantly upon mixing two solutions (darkening and precipitation of a product), which probably reflects less donor ability and thus less shielding of the coordination sphere by the latter solvents.

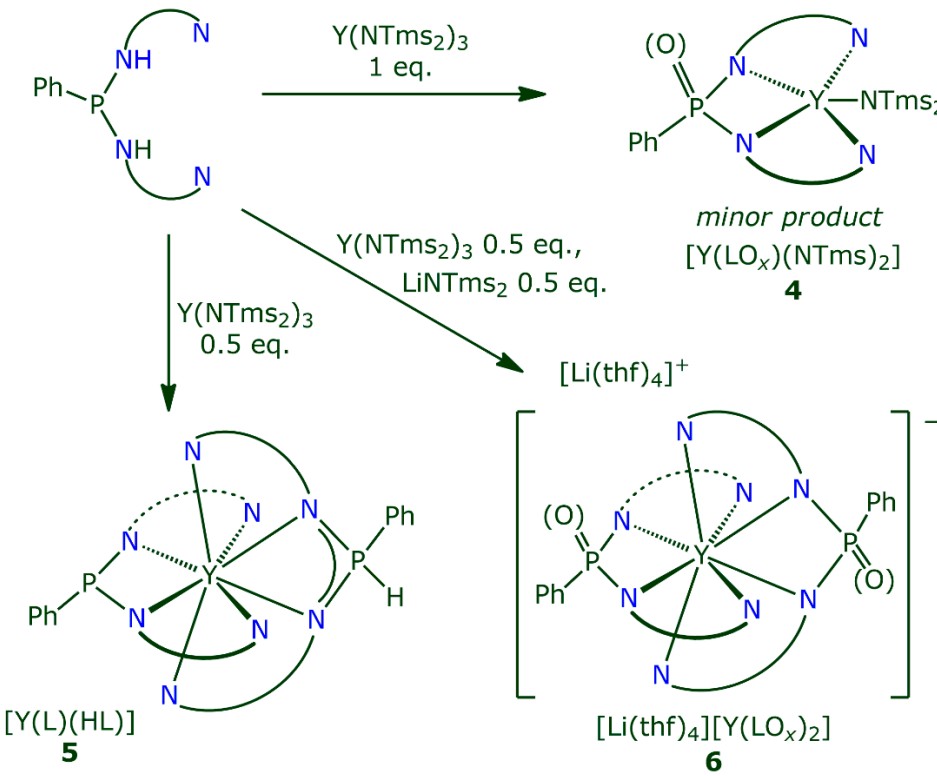

**Scheme 3.** Reactions of $H_2L$ with $Y(NTms_2)_3$. *Top:* in the ratio of 1:1, with the formation of $[Y(PhPO_x(NBtd)_2)NTms_2]$ (**4**). *Bottom left:* in the ratio L:Y = 2:1, with the formation of $[YL(HL)]$ (**5**). *Bottom right:* in the ratio L:Y:Li = 2:1:1, with the formation of $[Li(thf)_4][Y(PhPO_x(NBtd)_2)_2]$ (**6**).

Independent of the reaction setup, notable amounts of $Y(NTms_2)_3$ remained intact, which implies that not only the expected substitution of two amides and formation of $Y(L)(NTms_2)$ takes place. The obtained dark-brown product is apparently a mixture of some compounds, since it contained only broad and poorly resolved peaks in both $^1H$ and $^{31}P$ spectra. After a reaction in the variant of slow addition of $H_2L$ to Y amide was conducted, some amount of red-brown plates of the complex $[Y(PhPO_x(NBtd)_2)NTms_2]_2 \cdot Et_2O$ ($\mathbf{4}_2 \cdot Et_2O$) crystallized. The partly oxidized ligand is further abbreviated as $(LO_x)^-$. The structure of this compound, determined by XRD, comprises one tetradentate $L^{2-}$ and one $NTms_2^-$ ligand coordinated to the Y atom. There are two crystallographically independent $\{Y(LO_x)NTms_2\}$ units in the structure. Each of them is combined with its inversion equivalent in a head-to-tail mode (Figure 2). Two nearly flat $\{YLO_x\}$ fragments are placed antiparallel with the interplanar distance of ca. 3.4 Å, thus implying notable $\pi$-interactions in this dimer. An additional maximum of electronic density was found in the residual electronic density map near the phosphorus atoms, which was identified as a partially occupied oxygen atom ($d(P–O) = 1.567(19)$ and $1.736(14)$ Å), with the occupancy for two non-equivalent molecules of 0.13 and 0.22. This oxygen participates in the direct binding with the central atom of another half of the molecule. Alternatively, in the absence of the O atom, so does the P atom, forming a short $P \cdots Y$ contact ($d(P \cdots Y) = 2.875(3)$ and $2.993(3)$ Å). There is a disorder of the P atom over two proximate positions, likely caused by a slight difference in the geometry of the oxidised and non-oxidised species; this hampers the correct localization of the P atom and accurate determination of bond lengths. We explain the appearance of the O atom by ready oxidation of the ligand $L^{2-}$ by trace amounts of $O_2$, which could be present despite thorough control of the purity of the starting materials and accurate experiment conduction (in argon with typically less than 2 ppm of $O_2$, or in vacuum). Possibly, oxidized species are more prone to associate due to additional binding through the $\mu$-O atom in the dimer $\mathbf{4}_2$, and thus more readily crystallize. Despite our

attempts to deliberately obtain larger amounts of $\mathbf{4}_2$, its isolation was not successful, and its further spectral characterization as a single phase was not possible.

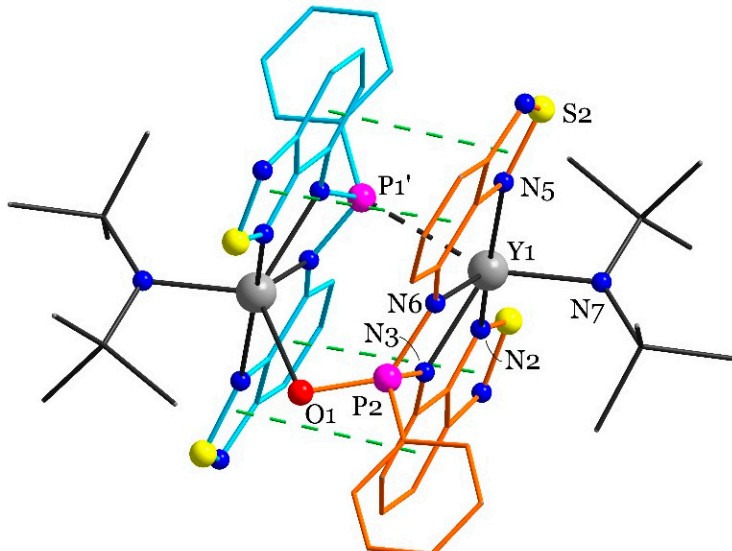

**Figure 2.** The structure of $\mathbf{4}_2 \cdot Et_2O$ according to XRD. Solvate ether molecules, disordered parts, and hydrogen atoms are omitted; one independent molecule is shown. Different L ligands are coloured with orange or blue. A partially occupied oxygen is located at both equivalent P atoms, but it is shown only at one of them. Green dashed lines indicate π-π interactions. Selected distances (Å): Y1–N2 2.545(3); Y1–N3 2.333(3); Y1–N5 2.580(3); Y1–N6 2.320(3); Y1–N7 2.221(4); Y2–N9 2.534(3); Y2–N10 2.377(3); Y2–N12 2.536(3); Y2–N13 2.491(3); Y2–N14 2.265(3). Symmetry code for ('): 1–*x*, –*y*, 3–*z*.

The interaction of two equivalents of $H_2L$ with $Y(NTms_2)_3$ (in toluene-ether mixture) proceeds fast upon mixing and leads to more definite outcomes (Scheme 3). A dark green-brown precipitate of the complex [YL(HL)] (**5**) is formed, which was separated as a single product after washing with ether and drying. The powdered product could be recrystallized from thf-hexane mixture (1:2 v/v) in the form of thin plate-like crystals, which were not suitable for XRD. In our hands, only crystallization in the presence of 1,4-dioxane (from its mixture with hexane) gave crystals in the form of large blocks appropriate for XRD. The crystal structure comprises both monoanionic $HL^-$ and dianionic $L^{2-}$ ligands surrounding one $Y^{3+}$ cation to form neutral {YL(HL)} units; both ligands are coordinated in tetradentate fashion (Figure 3a). Rather unexpectedly, two ligands do not saturate the coordination sphere, and leave enough space for coordination of an O atom of a dioxane molecule (coordination environment of the Y is thus {$ON_8$}). Coordinated dioxane bridges two crystallographically equivalent units, with the inversion centre in the middle of the molecule, thus leading to the formula [$\mathbf{5}_2(\mu\text{-dioxane})$]·dioxane. The second molecule of dioxane supports the packing of the molecules and is not coordinating. Protonated and not protonated form of the ligand can be distinguished by the lengthened Y–N(P) bonds and the shortened P–N bonds for $HL^-$ (by ca. 0.1 Å, see Table S1 and caption to Figure 3 for main bond distances); the latter are close to those in $\mathbf{2}_2 \cdot Et_2O$. In addition, residual electron density map for [$\mathbf{5}_2(\mu\text{-dioxane})$]·dioxane (Figure S1) allows for the explicit location of the hydrogen. In the $^{31}P$ NMR spectra of the powdered reaction product **5**, there are only two resonances with the ratio 1:1. The peak corresponding to $L^{2-}$ is found at typically lower fields (at 97 ppm), while that of $HL^-$ (at 9 ppm) is a doublet with large constant characteristic to a direct P–H bonds (457 Hz). In IR spectrum, there is the band of ν(PH) at 2275 cm$^{-1}$. Thus, analogously to the compound **2**, a fast NH to PH tautomeric shift occurs upon complexation with $HL^-$ ligand.

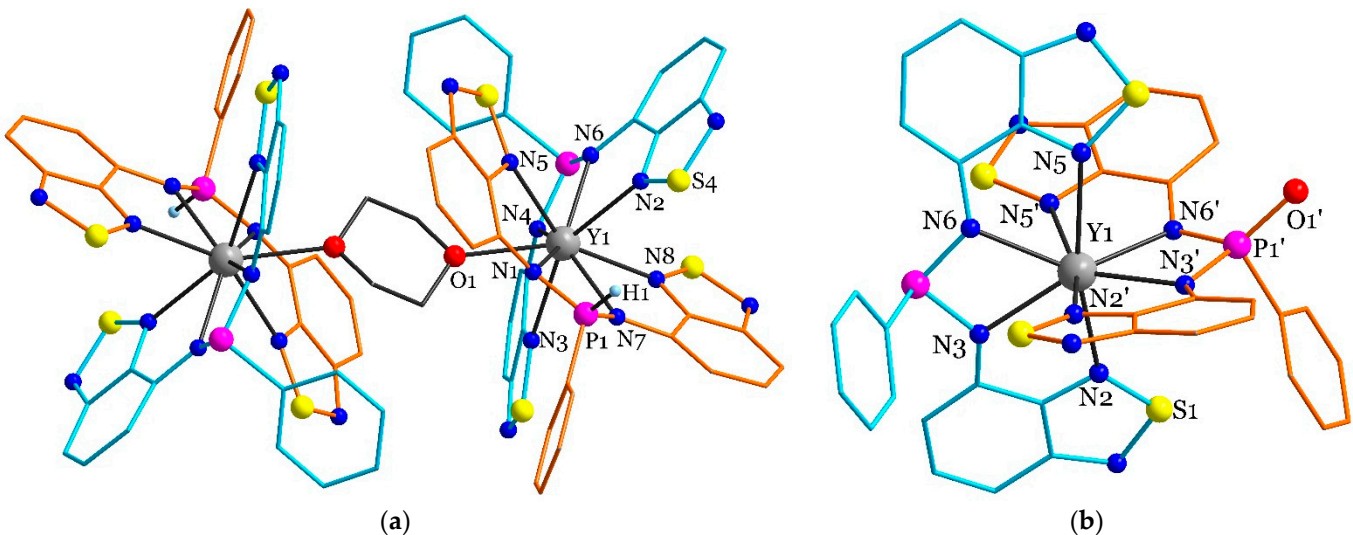

**Figure 3.** The structures of (**a**) [**5**$_2$·(μ-dioxane)]·dioxane and (**b**) the complex anion in **6**·hexane and **6**·1.5thf on the example of the oxygen-containing molecule of the former, according to XRD. Solvate molecules, the counterion, disordered parts, and hydrogen atoms are omitted. Different L/HL ligands are coloured with orange or blue. Selected distances for (**a**) (Å): Y1–N1 2.496(2); Y1–N2 2.631(2); Y1–N3 2.674(2); Y1–N4 2.388(2); Y1–N5 2.611(2); Y1–N6 2.372(2); Y1–N7 2.450(2); Y1–N8 2.540(2). Selected distances for (**b**) (Å): Y1–N2 2.582(4); Y1–N23 2.582(4); Y1–N33 2.371(4); Y1–N3 2.371(4); Y1–N5 2.544(4); Y1–N6 2.388(4). Symmetry code for ('): 1-*x*, *y*, 3/2-*z*.

The remaining PH proton in complex **5** can also be removed upon action of another equivalent of a base (Scheme 3). Thus, in the reaction of two equivalents of H$_2$L with one equivalent of Y(NTms$_2$)$_3$ and one equivalent of LiNTms$_2$, the compound [Li(thf)$_4$][Y(PhPO$_x$(NBtd)$_2$)$_2$] is cleanly formed, which could be crystallized from thf or hexane-thf mixtures as corresponding solvates, **6**·hexane, and **6**·1.5thf (Figure 3b). Both solvates, with one hexane or 1.5 thf solvate molecules per complex, contain the [Y(LO$_x$)$_2$]$^-$ anion of nearly the same geometry. The structure of this anion resembles the complex unit in [**5**$_2$(μ-dioxane)], with the exception for the absence of the dioxane ligand and the absence of the proton. The L$^{2-}$ ligands coordinate the Y in a tetradentate fashion, resulting in the coordination environment of {N$_8$}. Despite crystallization in the presence of thf, its molecules are not coordinated to the Y$^{3+}$ cation; this is likely owing to more close binding of dianionic ligands to the metal centre, than of monoanionic in **5**, thus leaving less vacant place in the coordination sphere. Indeed, the Y–N bonds in [YL$_2$]$^-$ are shortened by ca. 0.1 Å comparing to the corresponding bonds in **5** (cf. Table S1). Notably, **6**·hexane contains [YL$_2$]$^-$ anions that co-crystallize with a partially oxidized species, [Y(LO$_x$)$_2$]$^-$, occupying the same positions in the crystal. Analogously to **4**, an oxygen atom bonded to the phosphorus (d(P–O) = 1.716(13) Å) in two crystallographically equivalent ligands has the site occupancy *x* of 0.22. The change of the anionic shape, induced by the introduction of the oxygen, negligibly affects the crystal packing by a slight change of the position of one of thf molecules in the counterion [Li(thf)$_4$]$^+$. Thus, the ability of the oxidised and non-oxidised species to form a co-crystal is governed by a relatively loose packing of the species in the crystal. In the structure of **6**·1.5thf, no prominent electron density around the phosphorus, which corresponds to an oxygen atom, was observed.

In the compounds **4**$_2$·Et$_2$O, [**5**$_2$(μ-dioxane)]·dioxane, **6**·hexane, and **6**·1.5thf, the coordination mode of the L$^{2-}$ (or HL$^-$) ligands is similar. However, their conformation in the complexes with the two ligands differs from that in **4**. The latter reveals almost coplanar Btd moieties in the ligand: the dihedral angle between the RMS planes of the Btds does not exceed 10°. In the complexes [YL(HL)] and [YL$_2$]$^-$, Btd moieties belonging to one ligand are not coplanar, but inclined relative to each other with the dihedral angle varying in the range of 26.3–34.6°. The strongest deviation from the coplanarity is observed in

[**5**$_2$(μ-dioxane)], comprising an additional dioxane ligand in the complex. This is likely caused by a steric hindrance of two L$^{2-}$ (or HL$^-$) ligands in the coordination sphere of the bis-ligand complexes [YL(HL)] and [YL$_2$]$^-$, while the complex [YL{NTms$_2$}], having one L, shows the flattened arrangement of the Btds due to a less steric hindrance. The non-coplanarity of the Btds in [YL(HL)] and [YL$_2$]$^-$ is not caused by crystal packing effects, since the DFT-optimised in vacuum models (see below) reveal the same conformation of the ligands.

### 2.3. UV-Vis Spectroscopy and TD-DFT Calculations

Hereinafter, we will specify the forms of HL$^-$ with the proton riding at the N or P atom by the abbreviations H$^N$L$^-$ and H$^P$L$^-$. UV-Vis spectra of the compounds in thf solutions (ca. 10$^{-4}$ M) are presented in Figure 4. The position of the bands does not change upon dilution of the solutions five times (Figure S2). The shape of the spectra of [Li$_2$L] (**3**) and (Li(thf)$_4$)[YL$_2$]$^-$ (**6**), comprising the dianionic species L$^{2-}$, is quite similar. The compounds reveal poorly resolved bands spanning the entire visible region, which define the dark blue colour of the L$^{2-}$ species. The bands can be tentatively divided into short wavelength (400–470 nm) and long wavelength (470–750 nm) bands with the prevalence of the latter. In the case of compound [YL(HL)] (**5**) bearing both dianionic and monoanionic species, the spectrum resembles those for **3** and **6**, with the exception of somewhat more resolved bands that feature a clearer separation with the reduced intensity at ca. 520 nm (determining the green colour of **5**). In addition, the long wavelength band in **5** does not dominate compared to the short-wavelength band.

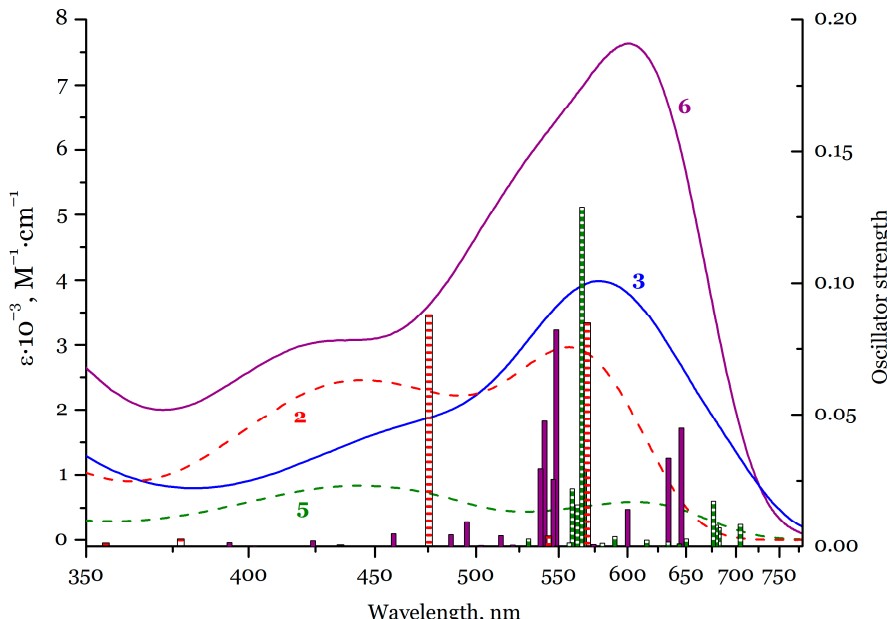

**Figure 4.** Experimental UV-Vis spectra of **2** (red dashed line), **3** (blue solid line), **5** (green dashed line), and **6** (violet solid line) in thf solutions (ca.·10$^{-4}$ M). Calculated electronic transitions of the compounds (vertical bars of the corresponding colours; TD-PBE0/def2-TZVPD level). Calculation for Li(thf)$_2$H$^N$L molecule is presented.

According to TD-DFT calculations for the single anion [YL$_2$]$^-$ at its ground-state optimised geometry (Table 1 and Table S2), the long wavelength band is composed of three nearly degenerate S$_0$–S$_n$ (*n* = 1–3) transitions, which correspond to the promotion from HOMO to LUMO (Figure 5). HOMO is predominately localised on two P and two Btd moieties, while LUMO is on Btd moieties of both ligands. This is manifested in a large charge transfer (CT); a similar behaviour (orbital distribution and strong absorption in the visible region) was observed in the zinc complex [ZnL$_2$] [27]. The binuclear molecule [{YL(HL)}$_2$(dioxane)], derived from the XRD analysis and optimised at its ground state,

also features first four CT transitions, corresponding to promotions from a combination of HOMO and HOMO–1 to LUMO and LUMO+$n$ ($n$ = 1–3) orbitals (Figure 5 and Figure S3, Table S3) of the same nature as for $[YL_2]^-$. Notably, the occupied orbitals are localised only on $L^{2-}$ ligands, i.e., they include P atoms that do not bear protons. The unoccupied orbitals are mainly localized on Btd moieties of the $HL^-$ ligands. Contrary to the $L^{2-}$ species, neutral $H_2L$ reveals only one hypsochromically shifted band in the visible region (Figure S2), featuring a lesser degree of CT character [27]. In the case of phases **4**, the presence of a very broad band (370–700 nm) in the UV-Vis spectrum (Figure S2) suggests a complicated mixture of species, in agreement with the NMR data.

**Table 1.** Experimental maxima of the long wavelength bands and calculated characteristics of the first vertical $S_0$–$S_n$ transitions for the compounds; $f$—oscillator strength, H—HOMO, L—LUMO.

| | λ(exp), nm | n | λ(calc), nm | f | Orbital Transitions * | Contribution |
|---|---|---|---|---|---|---|
| [Li(HL)] ** (**2**) | 560 | 1 | 570 | 0.085 | H → L | 0.069 |
| | | | | | H → L + 1 | 0.915 |
| | | 2 | 543 | 0.004 | H → L | 0.919 |
| | | | | | H → L + 1 | 0.066 |
| [YL(HL)] (**5**) | 605 | 1 | 705 | 0.009 | H → L | 0.900 |
| | | | | | H−1 → L + 1 | 0.060 |
| | | 2 | 682 | 0.007 | H−1 → L | 0.077 |
| | | | | | H−1 → L + 1 | 0.767 |
| | | 3 | 679 | 0.008 | H−1 → L + 1 | 0.112 |
| | | | | | H−1 → L + 3 | 0.839 |
| | | 4 | 677 | 0.017 | H−1 → L + 2 | 0.909 |
| [Li(thf)$_4$][YL$_2$] (**6**) | 600 | 1 | 646 | 0.045 | H → L | 0.958 |
| | | 2 | 634 | 0.034 | H → L | 0.963 |
| | | 3 | 600 | 0.014 | H → L | 0.933 |

*—only transitions with contributions greater than 0.05 are listed; **—calculation for Li(thf)$_2$H$^N$L molecule is presented.

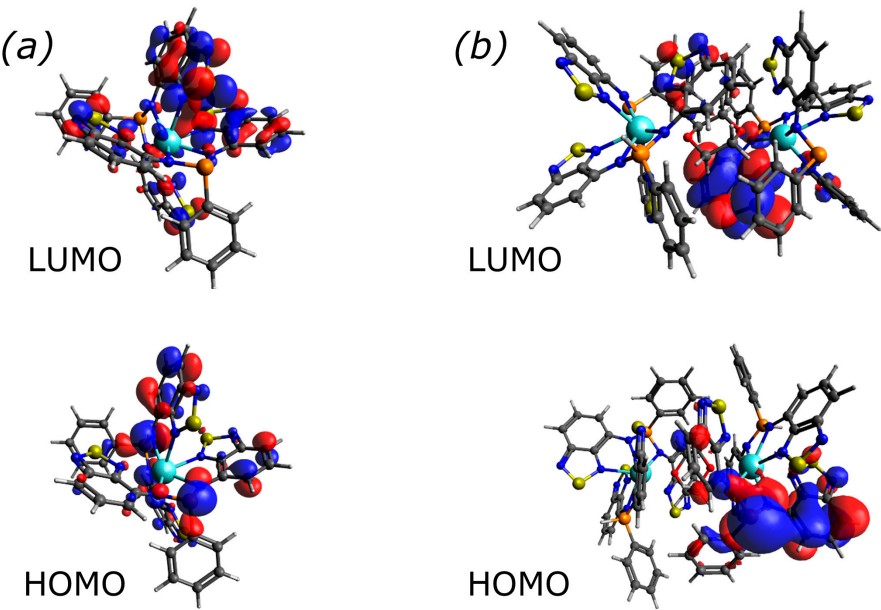

**Figure 5.** Frontier molecular orbitals for the complexes (**a**) $[YL_2]^-$ and (**b**) $[\{YL(HL)\}_2(dioxane)]$ at TD-PBE0/def2-TZVPD level (isovalue = 0.03).

For **2** comprising monoanionic species $HL^-$, the two bands were also observed in the visible region, with the distinctive long wavelength band slightly bathochromically shifted comparing to those for the other compounds. TD-DFT-computed transitions for the model $[Li(H^PL)]_2$, derived from the XRD, do not span the long wavelength region (the

first transition locates at 475 nm; Figure S4 and Table S4), which implies that the nature of the HL$^-$ species present in the solid is not the same as in the solution. A noticeably lighter colour of solid **2** compared to the solution, as well as the PH to NH proton transfer, was discussed above. Calculation of the transitions for a model molecule [Li(H$^N$L)(thf)$_2$], constructed with the proton at the amino group (Table S5, Figures S4 and S5), gives much better agreement with the experiment: the first calculated band locates at 570 nm. We can conclude that in the solid state, the H$^P$L$^-$ species with the proton at the phosphorus is more thermodynamically stable with respect to the H$^N$L$^-$ owing to the tetradentate coordination of the ligand. In thf solution of the yttrium complex **5**, the H$^P$L$^-$ species persists, since the relatively strong tetradentate coordination to the Y does not allow migration of the proton to the nitrogen. In the solution of **2**, the coordination is not that strong, while lithium prefers to include additional solvent thf molecules in its coordination sphere. As a result, N atom of one of the amino groups becomes capable for the attachment of the proton, which results in the proton migration and the formation of the H$^N$L$^-$ species.

## 3. Materials and Methods

### 3.1. General Methods

All manipulations were performed in strictly anaerobic conditions, in argon atmosphere (in a glove-box) or in vacuum. Schlenk vessels with greaseless PTFE plugs and flame-sealed ampoules were used for experiments. Solvents were dried and deoxygenated by distillation over Na-K alloy (with the addition of benzophenone for thf) and were condensed to the reaction mixtures in vacuum. Starting proligand H$_2$L was prepared according to the literature method [27] and recrystallized from a toluene-hexane mixture to ensure the absence of phosphine-oxide peaks in $^{31}$P NMR spectrum. Starting amide Y(NTms$_2$)$_3$ was prepared according to the literature method [37] and freshly sublimed in a vacuum before experiments; Li(NTms$_2$) was purchased (98%, Merck, Rahway, NJ, USA) and sublimed prior to the use. IR spectra were recorded by means of a FT-801 spectrometer (Simex, Novosibirsk, Russia) in KBr discs, which were prepared in the glove box and transferred in argon-filled vials. Elemental analysis for C, H, N, and S was performed by means of Vario Micro Cube analyzer (Elementar, Langenselbold, Germany). $^1$H NMR spectra (500.13 MHz) and $^{31}$P NMR spectra (202.45 MHz) were taken by means of a Bruker DRX-500 spectrometer (Bruker corporation, Billerica, MA, USA) in thf-d$_8$ at room temperature; the solvent peak was used as the internal reference for $^1$H spectra (1.72 ppm). Electronic absorption spectra were obtained by means of a Cary 60 UV-Vis spectrometer (Agilent, Santa Clara, CA, USA) equipped with a Xenon flash lamp (80 Hz) as the excitation source. Spectra were measured for the solutions in thf (two concentrations in the range $10^{-4}$–$10^{-5}$ M) in quartz cuvettes ($l$ = 1 cm) prepared by dissolution of exact weights of solid compounds; the graphs were scaled to molar extinction coefficients (see Figure 4). The solutions were prepared under inert atmosphere and flame-sealed in cuvettes with attached glass tubes to preclude ingress of air and oxidation or hydrolysis of the samples.

### 3.2. Quantum Chemical Calculations

All calculations were performed using Orca 5.0.3 quantum chemistry program suite [38]. Ground state geometries were optimized with PBE0 functional [39] and D3(BJ) dispersion correction [40]. Geometries of the anion [YL$_2$]$^-$ and the lithium salts [Li(thf)$_2$H$^N$L] and [LiH$^P$L]$_2$ were optimized using def2-SVPD basis set. Def2-SVP basis set was utilized for optimization of the binuclear complex [{Y(HL)L}$_2$(dioxane)] due to the large size of the molecule. The correspondence to the global minima was checked by analytical frequencies calculations, and no imaginary frequencies were found. The first 20 excitation energies were calculated for the optimized geometries by TD-DFT method with PBE0 functional and def2-TZVPD(-f) basis set. All calculations were accelerated by an RIJCOSX algorithm with corresponding auxiliary basis sets [41,42]. All calculations are performed for molecules in vacuum.

### 3.3. X-ray Structure Determination

Single-crystal XRD data for the compounds (Table S6) were collected at 150 K with a Bruker D8 Venture diffractometer with a CMOS PHOTON III detector and IμS 3.0 microfocus source (MoK$_\alpha$ radiation (λ = 0.71073 Å), collimating Montel mirrors). The crystal structures were solved using the SHELXT [43] and were refined using the SHELXL [44] programs with OLEX2 GUI [45]. Atomic displacement parameters for non-hydrogen atoms were refined anisotropically, with the exception of some disordered solvent molecules. The latter were refined with DFIX, DANG, ISOR, and RIGU restraints, as well as EADP constraints where needed. Hydrogen atoms were placed geometrically and refined in the riding model. For the structures of **4**$_2$·Et$_2$O and **6**·hexane, the maximum of electronic density near the phosphorus atoms was identified as an oxygen atom with partially occupied position, whose occupancy (SOF) was refined. Structure **4**$_2$·Et$_2$O revealed a disorder of the P atom over two proximate positions likely caused by slight differences in the geometry of the oxidised and non-oxidised species. The occupancy of the phosphorus was refined with the constraint of the sum of the SOFs equal to unity, and with equality of SOFs of the P and the riding O atoms. The structures were deposited to the Cambridge Crystallographic Data Centre (CCDC) as a supplementary publication, No. 2222413-2222417.

### 3.4. Syntheses

3.4.1. Synthesis of [Li(PhP(H)(NBtd)$_2$]$_2$·Et$_2$O (**2**$_2$·Et$_2$O)

To a mixture of H$_2$L (198 mg, 0.485 mmol) and Li(NTms$_2$) (82 mg, 0.49 mmol), prepared in the glovebox, 10 mL of thf was condensed upon cooling in liquid nitrogen. The mixture was left at room temperature with stirring. Upon warming, the colour changed from green (at low temperature) through dark-blue (close to ambient temperature) to reddish-violet (after several minutes at RT). The mixture was left with stirring overnight, after which it became dark cherry-red. The solution was evaporated to dryness in vacuum. To the obtained red-brown solid 10 mL of Et$_2$O was condensed. Upon stirring, the residue fully transformed into orange-red crystalline precipitate. The solution was removed with a pipet, the precipitate was washed with Et$_2$O and dried in vacuum. Single crystals suitable for XRD were picked directly from the crystalline batch. Yield was 134 mg (61%). Analysis found: C 53.2, H 3.9, N 18.3, S 14.8%; calc. for C$_{40}$H$_{34}$Li$_2$N$_{12}$OP$_2$S$_4$: C 53.21, H 3.80, N 18.62, S 14.21%. $^1$H NMR (δ, ppm): 7.64 (t, J = 6.0 Hz, 2H), 7.41 (br. s, $w_{1/2}$ = 32 Hz, 1H), 7.35–7.10 (m, 6H), 6.71 (br. s, $w_{1/2}$ = 24 Hz, 1H), 6.45 (br. s, $w_{1/2}$ = 22 Hz, 1H), 6.02 (s, 1H, N–H), 3.38 (q, Et$_2$O), 1.11 (t, Et$_2$O). $^{31}$P NMR (δ, ppm): 64.0 (s). IR (cm$^{-1}$): 650 w, 690 w, 705 w, 737 m, 766 w, 798 w, 841 w br, 901 m, 923 m, 953 m sh, 974 s, 1054 w, 1120 s, 1183 m, 1297 s, 1320 s, 1366 m, 1392 m, 1436 w, 1479 s, 1529 s, 1578 m, 2269 w br, 2864 w br, 2970 w, 3054 w, br.

3.4.2. Reaction of H$_2$L with Two Equivalents of Li(NTms$_2$)

To a mixture of H$_2$L (104 mg, 0.255 mmol) and Li(NTms$_2$) (85 mg, 0.51 mmol), 10 mL of thf were condensed upon cooling with liquid nitrogen. The mixture was left at room temperature with stirring. Upon warming, the colon changed to green and then to dark blue. The mixture was left with stirring overnight, then evaporated to dryness in a vacuum to give a dark-blue solid film on the walls. Diethyl ether (10 mL) was condensed to the solid residue. Upon sonication, the film was broken to thin powder, which did not look crystalline; the ether was only slightly coloured. The solvent was removed by a cannula, and the precipitate was dried in vacuum. Yield of the dried product (**3**) 102 mg. Analysis found: C 53.1, H 4.2, N 17.1, S 13.6%; calc. for C$_{20,4}$H$_{16,4}$Li$_2$N$_6$O$_{0,6}$PS$_2$ (Li$_2$L(thf)$_{0.3}$(Et$_2$O)$_{0.3}$, solvate composition according to NMR): C 52.79, H 3.56, N 18.1, S 13.82%. $^1$H NMR (δ, ppm): 7.66 (br. s, $w_{1/2}$ = 21 Hz, 2H), 7.25–6.95 (m, 6H), 6.41 (br. s, $w_{1/2}$ = 26 Hz, 3H), 3.61 (m, thf), 3.38 (q, Et$_2$O), 1.77 (m, thf), 1.11 (t, Et$_2$O). $^{31}$P NMR (δ, ppm): 77.2 (s), 70.4 (s). IR (cm$^{-1}$): 659 w br, 700 w, 740 m, 800 w, 824 w sh, 851 m br, 905 m, 971 w br, 1046 w br, 1095 s, 1116 m sh, 1179 w, 1286 s br, 1319 m, 1349 s, 1388 m, 1463 s br, 1525 m, 1560 m, 2878 w br, 2977 w, 3049 w br.

### 3.4.3. Reaction of H$_2$L with One Equivalent of Y(NTms$_2$)$_3$. Crystallization of [Y(PhPO$_x$(NBtd)$_2$)(NTms$_2$)]$_2$·Et$_2$O (**4**$_2$·Et$_2$O)

An ampoule with an additional section joined at the right angle was used for this reaction. Into the side section, H$_2$L (98 mg, 0.24 mmol) was placed; into another one, Y(NTms$_2$)$_3$ (140 mg, 0.246 mmol) was placed. Upon cooling in liquid nitrogen, 10 mL of Et$_2$O was carefully condensed only to the amide. The ether was frozen, and the ampoule was evacuated and flame-sealed. After thawing, the ampoule was placed with the amide-containing section nearly vertically above a heated plate (70 °C), so that upon warming the ether slowly condensed in the side section and washed the proligand into the reaction mixture. The complete dissolution of H$_2$L took ca. 2 days, the reaction mixture turned dark-green with the first portions of H$_2$L added, and with the time dark precipitate formed on the walls. After complete transfer of H$_2$L, the solvent was fully condensed to the emptied section. The soluble reaction products were then extracted by placing the ampoule in a temperature gradient ca. 50 °C–RT, over a course of one day. After concentration of the extract, two types of crystals formed. A notable amount of colourless needles were determined to be the starting amide, on the basis of their appearance, high solubility, and unit cell parameters data for one crystal close to those of Tb(NTms$_2$)$_3$ [46]. Along with them, a relatively small number of red plates crystallized, which had the structure of **4**$_2$·Et$_2$O. Due to the low amount of these crystals, they were not further analysed.

### 3.4.4. Synthesis of [Y(PhP(H)(NBtd)$_2$)(PhP(NBtd)$_2$)] (**5**)

To a mixture of H$_2$L (240 mg, 0.588 mmol) and Y(NTms$_2$)$_3$ (167 mg, 0.293 mmol) ca. 15 mL of Et$_2$O were condensed upon cooling with liquid nitrogen, and the mixture was left with stirring at RT. Upon warming, the initially obtained yellow solution gradually darkened, and a dark precipitate formed. To complete the transformation, the closed vessel with the mixture was left in a 50 °C bath overnight. Fine dark precipitate was filtered from a lighter-coloured greenish solution, washed with diethyl ether, and dried in a vacuum. Yield 214 mg (81%). Analysis found: C 47.7, H 2.8, N 18.6, S 14.4%; calc. for C$_{36}$H$_{23}$N$_{12}$P$_2$S$_4$Y: C 47.90, H 2.57, N 18.62, S 14.21%. $^1$H NMR (δ, ppm): 8.94 (d, J = 457.2 Hz, 1H, P–H), 8.40 (m, 2H), 7.95 (t, J = 6.4 Hz, 2H), 7.86 (m, 3H), 7.44 (t, J = 7.2 Hz, 2H), 7.40–7.25 (m, J = 7.4 Hz, 5H), 7.21 (br. s, 2H), 7.14 (br. s, 1H), 6.91 (br. s, 1H), 6.65–6.40 (m, 6H). $^{31}$P NMR (δ, ppm): 96.8 (s, 1P), 8.9 (dt, $^1$J$_{PH}$ = 457 Hz, $^3$J$_{PH}$ = 14 Hz, 1P). IR (cm$^{-1}$): 668 w br, 698 w br, 742 m, 799 w, 825 w, 851 w, 880 w, 901 m, 957 m, 1041 w, 1108 m, 1123 m, 1185 w, 1276 w sh, 1306 m, 1325 s, 1363 s, 1395 m, 1403 m, 1436 w, 1472 s, 1483 s, 1533 s br, 1566 m, 1585 m, 2275 w br, 3047 w br. Larger crystals suitable for XRD could only be obtained by recrystallization from 1,4-dioxane-hexane mixture (2:1 *v/v*) in a sealed ampoule in the form of dioxane-bridged dimer [**5**$_2$(μ-dioxane)]·dioxane.

### 3.4.5. Synthesis of [Li(thf)$_4$][Y(PhP(NBtd)$_2$)$_2$]·thf (**6**·thf)

In a glovebox, a mixture of Y(NTms$_2$)$_3$ (140 mg, 0.246 mmol) and Li(NTms$_2$) (40 mg, 0.24 mmol) was dissolved in 10 mL of diethyl ether, and H$_2$L (201 mg, 0.492 mmol) was separately dissolved in 20 mL of toluene. The solution of H$_2$L was slowly added (in the course of 15 min) to the stirred solution of the amides by a cannula. Blue coloration instantly appeared after the beginning; upon addition, the colour became more saturated, and at some point, dark precipitate started to fall. The reaction mixture was evaporated in vacuum to give a wet solid. Diethyl ether (10 mL) was added; after sonication, the bluish solution was removed by a cannula, and the powdered residue was dried in vacuum. The product was dissolved in 5 mL of thf, and to the resulting dark-blue solution, ca. 10 mL of hexane was slowly added with constant stirring. Precipitated compound was filtered and dried in vacuum. To obtain single crystals suitable for XRD, a small amount of the powder was recrystallized in a sealed tube from a thf-hexane mixture (10:1 v/v). Yield 182 mg (60%). Analysis found: C 52.0, H 4.8, N 13.5, S 10.2%; calc. for C$_{56}$H$_{62}$LiN$_{12}$O$_5$P$_2$S$_4$Y: C 52.99, H 4.92, N 13.24, S 10.11%. Somewhat low carbon content can be attributed to the formation of carbonate upon combustion. $^1$H NMR (δ, ppm): 7.85 (t, J = 6.0 Hz, 2H), 7.36 (t, J = 7.5 Hz,

2H), 7.27 (t, J = 7.5 Hz, 1H), 7.20 (t, J = 8.0 Hz, 1H), 7.09 (t, J = 8.0 Hz, 1H), 6.47 (d, J = 8.5 Hz, 1H), 6.27 (t, J = 7.5 Hz, 2H), 6.26 (d, J = 8.5 Hz, 1H), 3.61 (m, thf), 1.77 (m, thf). $^{31}$P NMR (δ, ppm): 91.3 (s), 46.7 (s, ca. 9% admixture of P=O form). IR (cm$^{-1}$): 589 w, 625 w, 657 w, 685 w, 697 m, 724 w sh, 743 s, 798 w, 826 m, 848 m, 867 w, 902 s, 1043 m, 1086 w sh, 1105 s, 1178 w br, 1276 m, 1302 m, 1326 m, 1359 s br, 1396 m, 1471 s, 1524 s, 1565 m, 2878 w, 2977 w br, 3049 w br.

## 4. Conclusions

The bis(amino)phosphane PhP(NHBtd)$_2$ bearing the heterocyclic 2,1,3-benzothiadiazol-4-yl (=Btd) substituents was deprotonated by strong amide bases. The reactions readily proceed with the formation of anionic species of two types, HL$^-$ and L$^{2-}$, coordinated as ligands to Li$^+$ and Y$^{3+}$ cations. All four chelating N atoms can participate in complexation, while the structure of the complex depends on the size and coordination preferences of the cation. Small tetrahedral Li$^+$ cations can be coordinated by two N-Btd moieties of two ligands, thus leading to binuclear bis-ligand complex. Larger Y$^{3+}$ cations are capable of coordinating all four chelating N atoms, thus leading to the mononuclear bis-ligand complexes with intersecting nearly planar ligands around the metal centre. These are the first characterized rare earth compounds with NPN-type ligands (P$^{III}$). The monoanionic form of the ligand undergoes a PH–NH tautomeric proton transfer, which can be controlled by the environment (solvent). The reversibility of this process is shown for the first time by anionic amine-amidophosphane species, and examples of Li and Y complexes allow one to suggest that chelating coordination facilitates the formation of the PH form. Complexes of Y with one bis(amido)phosphane ligand could not be obtained straightforwardly, possibly owing to high reactivity of the remaining silylamide ligand. UV-Vis absorption spectra show a bathochromic shift of main absorption bands in the visible region in the row H$_2$L–HL$^-$–L$^{2-}$. According to TD-DFT calculations, the main bands are associated with a charge transfer involving Btd orbitals, which is increased in the charged form of the ligand. As an unexpected feature, two complexes crystallized with partially oxidized central P$^V$ atom, assumingly bearing an oxygen atom. Its appearance, despite the strictly anaerobic reaction conditions, reflects readiness of oxidation of central P atom in the ligands. The fact of co-crystallization of complexes with P$^{III}$ and P$^V$(O)-centered ligands leads to an idea that the latter could be a more stable (and not prone to oxidation) alternative to the former while retaining all structural features. Complexes with this type of ligands, i.e., deprotonated forms of PR(O)(NHR)$_2$, are barely known in the transition metal chemistry and not known for rare earths and could be considered interesting objects for further research.

**Supplementary Materials:** The following supporting information can be downloaded at: https://www.mdpi.com/article/10.3390/inorganics10120263/s1, Figure S1: Residual electron density map for compounds **2**$_2$·Et$_2$O and [**5**$_2$(μ-dioxane)]·dioxane; Figure S2: UV-Vis spectra of concentrated and diluted solutions of the compounds; Figure S3: Frontier molecular orbitals for the complexes [{Y(HL)L}$_2$(dioxane)]; Figure S4: Comparison of the computed electronic transitions for the models [Li(thf)$_2$(H$^N$L)] and [Li(H$^P$L)]$_2$; Figure S5: Frontier molecular orbitals for [Li(thf)$_2$(H$^N$L)]; Figures S6–S15: NMR spectra of the compounds; Figures S16–S17: IR spectra of the compounds. Table S1: M–N (M = Li, Y) and P–N bonds (Å) in the compounds; Tables S2–S5: Vertical S$_0$–S$_n$ (n = 1–20) transitions of [YL$_2$]$^-$, [{Y(HL)L}$_2$(dioxane)], [Li(thf)$_2$(H$^N$L)] and [Li(H$^P$L)]$_2$; Table S6: Crystal data and structure refinement for the compounds.

**Author Contributions:** Conceptualization, N.A.P.; funding acquisition, S.N.K.; investigation, R.M.K., N.A.P. and T.S.S.; supervision, S.N.K.; visualization, R.M.K., N.A.P. and T.S.S.; writing—original draft, N.A.P. and T.S.S.; writing—review and editing, S.N.K. All authors have read and agreed to the published version of the manuscript.

**Funding:** This research was funded by the Russian Science Foundation (project No. 21-13-00287). The UV-Vis spectral study was supported by the Russian Ministry of Science and Higher Education (grant No. 121031700321-3 and 121031700313-8).

**Data Availability Statement:** Not applicable.

**Acknowledgments:** The authors are grateful to the Supercomputer Centre of Novosibirsk State University for computational resources.

**Conflicts of Interest:** The authors declare no conflict of interest.

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
