# Peer review of "Yttrium and Lithium Complexes with Diamidophosphane Ligand Bearing 2,1,3-Benzothiazolyl Substituent: Polydentate Complexation and Reversible NH–PH Tautomery"

_inorganics, doi:10.3390/inorganics10120263_

Round 1

Reviewer 1 Report

Pushkarevsky,  Sukhikh et al. systematically study the coordination chemistry of Li and Y ions with a NPN-type ligand PhP(NHBtd)2 that can exist in HL- and L2- forms. Complexes [Li(HL)]2·Et2O (2), [Y(PhPOx(NBtd)2)NTms2]2·Et2O (4), [YL(HL)]2(µ-dioxane)]·dioxane (5), [Li(thf)4][YL2]·hexane (6·hexane), and [Li(thf)4][YL2]·1.5thf (6·1.5thf) were characterized by SCXRD, NMR, and UV-Vis spectroscopy, and the results were supported by TD-DFT calculations. Interestingly, a reversible NHPH tautomery of [Li(HL)]2 in the solid state was observed and discussed its solution behavior was studied by IR, NMR, UV-Vis spectroscopy, and TD-DFT calculations. Moreover, partially oxidized ligand PhPOx(NBtd)2 was observed in [Y(PhPOx(NBtd)2)NTms2]2 and [Li(thf)4][YL2]·hexane complexes. This work is systematic, detailed, and should be suitable for inorganics. The following suggestions for revision are made in the spirit of further enhancing teh quality of the work:  

  1. In the manuscript, NMR and IR are used to confirm the NH and PH configuration. It could be helpful to show them in SI.
  2. Both [Y(PhPOx(NBtd)2)NTms2]2·Et2O (4) and  [Li(thf)4][YL2]·hexane (6·hexane) comprise the partially oxidized ligand PhPOx(NBtd)2. Why is L used in the formula of 6, but PhPOx(NBtd)2 in 4?
  3. Please use a consistent abbreviation, either Btd or btd.
  4. Based on the facts that complex 6 shows long wavelength (470750 nm) absorption bands with no H atom close to the center Y ion, its analogs of Yb, Er, or Nd may show near infrared emission. It would be worthwhile to show this unique trait in one of such complexes.   

Author Response

1. In the manuscript, NMR and IR are used to confirm the NH and PH configuration. It could be helpful to show them in SI.

Yes, we agree; the NMR and IR spectra were added to the SI.

2. Both [Y(PhPOx(NBtd)2)NTms2]2·Et2O (4) and  [Li(thf)4][YL2]·hexane (6·hexane) comprise the partially oxidized ligand PhPOx(NBtd)2. Why is L used in the formula of 6, but PhPOx(NBtd)2 in 4?

This is a good point, thank you. In both compounds, the oxidized species occupy a minor part of positions in the crystal structure, so technically, most part of compound consists of non-oxidized molecules. We slightly changed the text so that (LOx) is used for both compounds 4 and 6. There were two structures of 6, crystallized independently, with thf and hexane solvate molecules. Only the structure with hexane contains oxidized form of the ligand, while the solvate with thf does not, which is also reflected in the text.

3. Please use a consistent abbreviation, either Btd or btd.

We have uniformed the abbreviation.

4. Based on the facts that complex 6 shows long wavelength (470750 nm) absorption bands with no H atom close to the center Y ion, its analogs of Yb, Er, or Nd may show near infrared emission. It would be worthwhile to show this unique trait in one of such complexes.

Thank you for the comment. The luminescence study would indeed provide an interesting results, especially in view of the comparison between the lanthanide species with and without the H atom at the phosphorus. We have already started the work on lanthanide analogues, but we cannot study their NIR luminescent properties at the moment. We anticipate this will be the matter of future research

Reviewer 2 Report

This article reports a series of yttrium and lithium coordination compounds constructed from heterocyclic organic ligands. The work is solid and the manuscript has been prepared with care. I like to see I t published in this journal after minor revisions.

1.       I suggest Table 1 moves to SI file with important bond length and pi-pi interaction etc. can be marked in structural Figures. Some atoms in Figures should be labelled as well. Also, a table for crystallographic data may help. Why saying “The flat Btd and Ph fragments of different ligands form three pairs of π-stacks.”, it is better to present a relevant figure to demonstrate the pi-pi interactions and the corresponding data be presented as well; may this can be done on Figure 1.

2.       More details for structural refinements and descriptions should be presented in main and Supporting information. For example, there are disorder and partial occupation of O of O-P in some structures. How were they dealt with? These should be well documented. And what are exactly the solvent molecules in the crystal structures? Et2O or CH3(CH2)4CH3? Everything should be consistent. The SOF for the partially occupied O should be identical in main text and CIF files.

3.       The compound names (abbreviations), structural formulas, etc should be clearly provided. Otherwise, it is hard to follow what the authors are discussing.

4.        Grammar errors and formatting problems.

1)       Please check the references formats for refs. 27, 36 and so forth to satisfy the requirements of the journal.

2)       THF rather than thf?

3)       Full name for thf should be mentioned in Abstract. What is “BTd” referred to? Any abbreviation should be explained at its first appearance.

4)       “i.e.” should also be italic.

5)       x, y, z, in symmetry codes should be italic. “e. g. halide” should be “e. g., halide”. Similar case for “(i. e. L2–).”.

6)      In the title, the “polydentate” should be “Polydentate”.

7)      Many sentences need to be polished. Such as “Hence, in this work we used similar approach, the reactions of H2L with readily available amides LiNTms2 and Y(NTms2)3, with an intent that the only by-product, amine HNTms2, is well soluble and could be easily separated, as opposed to precipitates of alkali halides formed in salt metathesis reactions.”, “It was of further interest, how the analogous bis(amido)phosphane(III) ligands (type B) will support the rare earth cations.”, “the mixture slowly darkened at RT “, “Interestingly, that there is an additional maximum of electronic density near the phosphorus atoms,”.

8)      “twice deprotonated” or “doubly-deprotonated.”.

9)      Standard deviation of bond distances like that in “(d(P–O) = 1.57 and 1.74 Å ),”.

Author Response

  1. I suggest Table 1 moves to SI file with important bond length and pi-pi interaction etc. can be marked in structural Figures. Some atoms in Figures should be labelled as well. Also, a table for crystallographic data may help. Why saying “The flat Btd and Ph fragments of different ligands form three pairs of π-stacks.”, it is better to present a relevant figure to demonstrate the pi-pi interactions and the corresponding data be presented as well; may this can be done on Figure 1.

We have revised Figures 1, 2 and 3 by labelling some atoms and have clarified the bond distances with the metals in the figure captions. In addition, we highlighted π-π interactions mentioned in the text.

  1. More details for structural refinements and descriptions should be presented in main and Supporting information. For example, there are disorder and partial occupation of O of O-P in some structures. How were they dealt with? These should be well documented. And what are exactly the solvent molecules in the crystal structures? Et2O or CH3(CH2)4CH3? Everything should be consistent. The SOF for the partially occupied O should be identical in main text and CIF files.

For the structures of 42·Et2O and 6·hexane, the maximum of electronic density near the phosphorus atoms was identified as an oxygen atom with partially occupied position, whose occupancy (SOF) was refined. Structure 42·Et2O revealed a disorder of the P atom over two proximate positions likely caused by a slight differences in the geometry of the oxidised and non-oxidised species. The occupancy of the phosphorus was refined with the constraint of the sum of the SOFs equal to unity, and with equality of SOFs of the P and the riding O atoms. We have included this information in the Experimental section, as well as clarified some restraints and constraints used to refine the solvent molecules. We have revised the text clarifying the exact SOFs for the O atoms.

The XRD analysis revealed the following formula of the compounds: 22·Et2O, 52(dioxane)·dioxane, 42·Et2O, 6·hexane, and 6·1.5thf. When discussing the crystal structures, we use these formula. In the manuscript, a molecule or ion pair is denoted by a number (e. g. 2), while the subscript denotes a dimeric species (e. g. 22). An additional donor directly bound to a molecule is specified in brackets (e. g. 2(thf)), while for a crystalline phase comprising a solvate molecule, the latter is specified after the dot (e. g. 22·Et2O). We have clarified this in the text.

  1. The compound names (abbreviations), structural formulas, etc should be clearly provided. Otherwise, it is hard to follow what the authors are discussing.

Thank you. We checked that the abbreviations such as Btd, L, structural formulas are provided at the first place of their appearance. We added several sentences defining the not so evident abbreviations (LOx)– (for partly oxidized ligands) and HNL/HPL (for the isomeric ligands with N- or P-bonded hydrogen atom, used for the calculations). In addition, we added the short formulas used in the text, to the schemes (such as [YL2]–). We hope that these improvements will be enough for following the discussion.

  1. Grammar errors and formatting problems.

1)       Please check the references formats for refs. 27, 36 and so forth to satisfy the requirements of the journal.

The references were checked and corrected.

2)       THF rather than thf?

We prefer the latter variant, which is also typically used in publications.

3)       Full name for thf should be mentioned in Abstract. What is “BTd” referred to? Any abbreviation should be explained at its first appearance.

We have clarified the abbreviation for thf. Btd is 2,1,3-benzothiadiazol-4-yl as mentioned in the abstract (at its first appearance) and in the introduction. In addition, Btd is represented in Scheme 1.

4)       “i.e.” should also be italic.

We have revised the text accordingly

5)       x, y, z, in symmetry codes should be italic. “e. g. halide” should be “e. g., halide”. Similar case for “(i. e. L2–).”.

We have revised the text accordingly

6)      In the title, the “polydentate” should be “Polydentate”.

We have revised the title accordingly

7)      Many sentences need to be polished. Such as “Hence, in this work we used similar approach, the reactions of H2L with readily available amides LiNTms2 and Y(NTms2)3, with an intent that the only by-product, amine HNTms2, is well soluble and could be easily separated, as opposed to precipitates of alkali halides formed in salt metathesis reactions.”, “It was of further interest, how the analogous bis(amido)phosphane(III) ligands (type B) will support the rare earth cations.”, “the mixture slowly darkened at RT “, “Interestingly, that there is an additional maximum of electronic density near the phosphorus atoms,”.

Thank you for defining the odd sentences. We checked them and tried to split or reformulate.

8)      “twice deprotonated” or “doubly-deprotonated.”.

We have revised the text accordingly

9)      Standard deviation of bond distances like that in “(d(P–O) = 1.57 and 1.74 Å ),”.

We have clarified the standard deviations (uncertainties) for the bond distances discussed in the text.

Reviewer 3 Report

This manuscript is of the correct length and scope for Inorganics. This is a clear and well-written paper and I recommend publication following the few minor points listed below.

Page 4, line 4: "... which means that ..." is probably best stated as "... suggests that ...".

Page 4, line 18: "... several oligomeric species ..."; perhaps reference of this type of observation might be prudent (e.g., van Koten, Dalton Trans 2008, 5783).

Page 8, line 7: "phosphorus" (sp.).

Standard journal abbreviation titles should be used in the reference section, e.g. 'Coordination Chemistry Reviews' as 'Coord. Chem. Rev.'.

Is ref. [36] complete?   

Author Response

1. Page 4, line 4: "... which means that ..." is probably best stated as "... suggests that ...".

We have revised the text accordingly

2. Page 4, line 18: "... several oligomeric species ..."; perhaps reference of this type of observation might be prudent (e.g., van Koten, Dalton Trans 2008, 5783).

 Thank you, this reference is indeed quite relevant.

3. Page 8, line 7: "phosphorus" (sp.).

We have revised the text accordingly

4. Standard journal abbreviation titles should be used in the reference section, e.g. 'Coordination Chemistry Reviews' as 'Coord. Chem. Rev.'.

Corrected for all references. 

5. Is ref. [36] complete?   

This reference (for Herrmann-Braues “Synthetic Methods…”) was corrected.